# Comparison and Functional Analysis of Odorant-Binding Proteins and Chemosensory Proteins in Two Closely Related Thrips Species, *Frankliniella occidentalis* and *Frankliniella intonsa* (Thysanoptera: Thripidae) Based on Antennal Transcriptome Analysis

**DOI:** 10.3390/ijms232213900

**Published:** 2022-11-11

**Authors:** Xiaowei Li, Jianghui Cheng, Limin Chen, Jun Huang, Zhijun Zhang, Jinming Zhang, Xiaoyun Ren, Muhammad Hafeez, Shuxing Zhou, Wanying Dong, Yaobin Lu

**Affiliations:** State Key Laboratory for Managing Biotic and Chemical Threats to Quality and Safety of Agro-Products, Key Laboratory of Biotechnology in Plant Protection, Ministry of Agriculture and Rural Affairs, Institute of Plant Protection and Microbiology, Zhejiang Academy of Agricultural Sciences, Hangzhou 310021, China

**Keywords:** *Frankliniella occidentalis*, *Frankliniella intonsa*, odorant-binding proteins, chemosensory proteins, aggregation pheromone, molecular docking

## Abstract

Two closely related thrips species, *Frankliniella occidentalis* and *Frankliniella intonsa*, are important pests on agricultural and horticultural crops. They have several similarities, including occurrence patterns, host range, and aggregation pheromone compounds. However, there are very few reports about the chemosensory genes and olfactory mechanisms in these two species. To expand our knowledge of the thrips chemosensory system, we conducted antennal transcriptome analysis of two thrips species, and identified seven odorant-binding proteins (OBPs) and eight chemosensory proteins (CSPs) in *F. occidentalis*, as well as six OBPs and six CSPs in *F. intonsa*. OBPs and CSPs showed high sequence identity between the two thrips species. The RT-qPCR results showed that the orthologous genes *FoccOBP1/3/4/5/6*, *FintOBP1/3/4/6*, *FoccCSP1/2/3*, and *FintCSP1/2* were highly expressed in male adults. Molecular docking results suggested that orthologous pairs *FoccOBP4*/*FintOBP4*, *FoccOBP6*/*FintOBP6*, and *FoccCSP2*/*FintCSP2* might be involved in transporting the major aggregation pheromone compound neryl (*S*)-2-methylbutanoate, while orthologous pairs *FoccOBP6*/*FintOBP6*, *FoccCSP2*/*FintCSP2*, and *FoccCSP3*/*FintCSP3* might be involved in transporting the minor aggregation pheromone compound (*R*)-lavandulyl acetate. These results will provide a fundamental basis for understanding the molecular mechanisms of pheromone reception in the two thrips species.

## 1. Introduction

Semiochemicals, such as plant volatiles and insect-produced pheromones, are known to produce a wide range of behavioral responses in insects, including host locating, mating, oviposition site selecting, avoiding natural enemies, etc. [1,2,3]. Such behaviors primarily rely on the precise detection and recognition of different semiochemicals through the olfactory system [4]. Consequently, studies of the chemosensory mechanisms are key to a better understanding of insect intra- and inter-specific interactions, insect–plant interactions, and insect environmental adaptation [5,6,7,8].

There are several kinds of olfactory proteins involved in the recognition of semiochemicals in insects: (1) odorant-binding proteins (OBPs) [9] and chemosensory proteins (CSPs) [10], two types of small soluble proteins that can bind and transport various odorant signals from antennal sensillar lymph to corresponding chemosensory receptors (CRs); (2) chemosensory receptors (CRs) [including odorant receptors (ORs), gustatory receptors (GRs) and ionotropic receptors (IRs)], which are on the dendrites of neurons, can be activated by odorant signals, turn odorant signals into electrical signals and stimulate insects to make behavioral responses accordingly [11]; (3) sensory neuron membrane proteins (SNMPs), which are essential for reception of fatty acid-derived odorants and pheromones in certain groups of insects [4,12]; and (4) odorant-degrading enzymes (ODEs), which could degrade odorants [4].

Among all these olfactory proteins, OBPs and CSPs are involved in the initial step of olfactory recognition [13]. OBPs and CSPs are two families of acidic, soluble proteins with a similar structure that binds to small organic compounds [14]. They were reported to be involved in the detection, identification and transportation of odorants and pheromones [14]. Classical OBPs are usually around 130~140 amino acid residues long and have six conserved cysteines paired into three interlocked disulfide bridges [15]. Most insect OBPs are made of a fold of six α-helical domains, forming a compact and stable structure [16]. The family of OBPs also includes non-classical ones with various numbers of conserved cysteines, such as Minus-C OBPs (4 conserved cysteines), Plus-C OBPs (8 conserved cysteines), dimer OBPs (12 conserved cysteines) and Atypical OBPs (9~10 conserved cysteines) [17,18,19,20]. While CSPs are usually smaller than OBPs (100~120 residues), they typically have four conserved cysteines forming two independent loops [21]. CSPs in insects are also made of α-helical domains but with different folding from OBPs [22,23,24]. Despite their structural differences, the two classes of soluble proteins are with hydrophobic binding pockets and have similar roles across species and orders [13].

Knowledge of the three-dimensional structures and binding modes could help in understanding the roles of the soluble olfactory proteins in insects [23]. To date, the structures of more than 20 OBPs have been reported in insects [25,26]. By contrast, the structures of very few CSPs are currently available [22,23,24,27]. The three-dimensional structure-based action mechanism of OBPs has been reported in insect species in Lepidoptera, Diptera, and Hemiptera, etc. [25]. However, ligand binding by OBPs seems to vary among insect species, presenting specific features depending on the species studied [25]. Molecular docking is a widely used virtual method for the prediction of the best ligands for insect OBPs [28], which has been also used for ligand binding prediction in CSPs [29]. This computational method is becoming a very important tool in screening for potential semiochemicals of given soluble olfactory proteins, as well as potential target soluble olfactory proteins of given ligands [30,31].

It has been reported that comparative analysis of chemosensory genes (OBPs, CSPs, or ORs) among closely related species could help understand the adaptation of insects to different ecological niches and their evolutionary relationships [8,29,32,33]. In two closely related *Anastrepha* fruit flies, *A. fraterculus* and *A. obliqua*, a comparison of OBP genes between the two species revealed four genes associated with positive selection which might precede the speciation of these two species; in addition, several amino acid changes between homologous genes might be related to different host preferences [8]. Comparison and functional analysis of CSP genes from two closely related weevils, *Eucryptorrhynchus scrobiculatus* and *E. brandti*, found candidate genes that might explain the niche differentiation of the two weevils [29]. Analysis of *Drosophila suzukii* chemoreceptors together with nine melanogaster group members, including two of its closely related species (*D. biarmipes* and *D. takahashii*), revealed several candidate receptors associated with the ecological niche adaptation of *D. suzukii* [32]. Comparison sequence analysis of pheromone receptors (PRs) between two closely related moth species, *Helicoverpa assulta* and *H. armigera*, indicated that species-specific changes in the tuning specificity of the PRs could be achieved with a few amino acid substitutions, which might shift the ligand selectivity of the PRs and thus trigger speciation of closely related moth species [33].

The closely related thrips species, *Frankliniella occidentalis* (Pergande) and *Frankliniella intonsa* (Trybom), are important pests on agricultural and horticultural crops. The former is a worldwide invasive species originally from North America, invaded China in 2003, and has now become one of the most significant horticultural pests in China [34], whereas the latter is a local widely distributed pest in the Palaearctic and Oriental regions, and now has expanded its range to Nearctic ecozones [35]. Both two species are highly polyphagous insects, causing damage to crops through feeding and egg-laying. Moreover, *F. occidentalis* and *F. intonsa* are important vectors of tospoviruses such as tomato spot wilt virus (TSWV) and Chrysanthemum stem necrosis virus (CSNV), resulting in destructive damage to crops [36,37]. As two closely related species, *F. occidentalis* and *F. intonsa* have similarities in several aspects. For instance, the two thrips are both flower-dwelling species, with a high feeding preference for flowers compared to other plant parts [38,39,40]. In addition, both of them have very wide host ranges and can co-occur on various vegetable plants [38,41]. *F. occidentalis* and *F. intonsa* also share two compounds, (*R*)-lavandulyl acetate and neryl (*S*)-2-methylbutanoate, as their aggregation pheromone compounds, but in different ratios, 1:12.949 and 1:2.722, respectively [42]. Field-trapping experiments showed that the optimal blends of synthetic (*R*)-lavandulyl acetate and neryl (*S*)-2-methylbutanoate were 1:8 (1250 ng: 10,000 ng) in *F. occidentalis* and 1:4 (1250 ng: 5000 ng) in *F. intonsa*, respectively [43].

Due to similar niches, competitive interaction between *F. occidentalis* and *F. intonsa* has been frequently reported. It has been reported that the relative abundances between *F. occidentalis* and *F. intonsa* varies depending on plant hosts and abiotic conditions [44,45,46,47,48,49]. Our previous study found that species-specific aggregation pheromones regulated by the ratio of pheromone compounds might also contribute to the coexistence of the two thrips species [50]. Given the overlap in pheromone compounds and niche similarities, it is reasonable to hypothesize that the olfactory system and underlying mechanisms might play an important role in interspecific interactions between the two closely related thrips species. However, there are very few reports about the chemosensory genes and olfactory mechanisms in these two species. In this study, the antennal transcriptome analysis was performed for *F. occidentalis* and *F. intonsa*, and the putative chemosensory genes (OBPs and CSPs) were identified and compared between the two species. Moreover, the candidate OBPs or CSPs involved in aggregation pheromone reception were predicted by molecular docking, which may provide a basis for understanding the molecular mechanisms of pheromone-mediated interspecific interaction between the two closely related species.

## 2. Results

### 2.1. Transcriptome Sequencing and Assembly

The six cDNA libraries (WFTF1, WFTF2, WFTF3, FTF1, FTF2, FTF3) were sequenced using Illumina Novaseq™ 6000, and over 49.05 Gb clean reads were obtained, with at least 5.37 Gb from each sample. The clean read number, base number, GC content, and Q30 (%) of all groups are shown in the Appendix A. A total of 48,889 unigenes were obtained after assembly, with the GC content 44.56% and the N50 value 1180 bp. Additionally, 74,380 transcripts were obtained, with the GC content 45.31% and the N50 value 1257 bp (Appendix A). The datasets of the transcriptomes from our study were uploaded to NCBI Sequence Read Archive with accession number GSE213075.

### 2.2. Annotation Information for Unigenes

Unigene functional annotation was performed in comparison with the Gene Ontology (GO), KEGG, Pfam, Swissport, eggNOG, and NR databases. A total of 48,889 unigene annotations were obtained: 18,484 in GO, 11,111 in KEGG, 18,772 in Pfam, 16,468 in Swissprot, 22,490 in eggNOG, and 21,362 in NR (Appendix A). Compared with the GO database, the thrips unigenes were divided into three categories: molecular function, cellular component, and biological process (Figure 1). In the molecular function category, nine unigenes showed olfactory receptor activity (GO:0004984), and one unigene had odorant-binding activity (GO:0005549). In the cellular component category, 4 unigenes were annotated to signal recognition particles (GO:0048500), while in the biological process category, 84 unigenes were involved in the Gprotein coupled receptor signaling pathway (GO:0007186), 30 unigenes participated in the sensory perception of smell (GO:0007608), and 16 unigenes were related to the detection of chemical stimulus involved in the senses (GO:0050911). These unigenes may have functions in olfactory perception, odorant detection, and host location (Figure 1). Based on the eggNOG database, 22,490 unigenes were categorized into 23 molecular families, of which 1361 unigenes were related to intracellular trafficking, secretion, and vesicular transport, and 808 unigenes were involved in signal transduction (Figure 2).

### 2.3. Identification and Analysis of Putative OBPs

Antennal transcriptome analyses of *F. occidentalis* and *F. intonsa* identified 13 different unigenes encoding putative OBPs, of which 11 had full-length open reading frames (ORFs) (Table 1). Among the 13 putative OBPs, 7 were OBP genes in *F. occidentalis* (FoccOBP1~7), and 6 were OBP genes in *F. intonsa* (FintOBP1~6). Among the 7 OBPs in *F. occidentalis*, FoccOBP1 (KM527948.1), FoccOBP2 (KM527950.1), and FoccOBP4 (JF937664.1) have been reported previously. FoccOBP2 and FintOBP2 belong to the Plus-C subfamily, while FoccOBP5 and FintOBP5 belong to the Minus-C subfamily. The remaining nine genes were classical OBPs. The alignment of candidate OBPs revealed that six orthologous pairs shared high amino acid identities (≥76.72%) between *F. occidentalis* and *F. intonsa*, respectively (Table 2; Figure 3A).

### 2.4. Identification and Analysis of Putative CSPs

Antennal transcriptome analyses of *F. occidentalis* and *F. intonsa* identified 14 different unigenes encoding putative CSPs, of which all had full-length open reading frames (ORFs) (Table 1). Among the 14 putative CSPs, 8 were CSP genes in *F. occidentalis* (FoccCSP1~8), and 6 were CSP genes in *F. intonsa* (FintCSP1~6). Among the eight CSPs in *F. occidentalis*, FoccCSP1 (KM035415.1), FoccCSP2 (JF937663.1), and FoccCSP6 (KM527949.1) has been reported previously. Among the seven CSPs in *F. intonsa*, FintCSP1 (MT211602.1) and FintCSP2 (MT199111.1) have been reported previously in GenBank. The alignment of candidate CSPs revealed that six orthologous pairs shared high amino acid identities (≥83.87%) between *F. occidentalis* and *F. intonsa*, respectively (Table 2; Figure 3B).

### 2.5. Phylogenetic Analysis of OBPs and CSPs in F. occidentalis and F. intonsa

For phylogenetic analysis of OBPs, a phylogenetic tree was established based on 167 OBPs of eight species from different orders: Orthoptera (*Locusta migratoria*), Hemiptera (*Acyrthosiphon pisum*), Diptera (Drosophila melanogaster), Coleoptera (*Tribolium castaneum*), Hymenoptera (*Apis mellifera*), Lepidoptera (*Spodoptera exigua*) and Thysanoptera (*F. occidentalis* and *F. intonsa*) (Appendix A). Four sub-groups have been identified on the tree (Figure 4). OBPs from *L. migratoria*, *D. melanogaster*, *T. castaneum* and *S. exigua* were distributed in all four sub-groups. Sub-group I was dominated by OBPs from *D. melanogaster* and *S. exigua*. Sub-group II mainly included OBPs from *S. exigua* and *A. pisum*, but no *A. mellifera* OBPs clustered in this subgroup. Sub-group III included OBPs from all eight species, including the two species in our study, *F. occidentalis* and *F. intonsa*. Sub-group IV was dominated by OBPs from A. mellifera, S. exigua and D. melanogaster, but no A. pisum OBPs clustered in this sub-group. In addition, two OBPs from both *F. occidentalis* and *F. intonsa* were distributed in sub-group IV. Based on the phylogenetic relationship, FoccOBP7 was a homolog of ApisOBP4, FoccOBP1/FintOBP1 homologs were closely related to SexiOBP21~24, FoccOBP2/FintOBP2 homologs were closely related to SexiOBP36, FoccOBP3/FintOBP3 and FoccOBP6/FintOBP6 homologs were closely related to SexiOBP14 and TcasOBP16, FoccOBP4/FintOBP4 homologs were closely related to SexiOBP38, and FoccOBP5/FintOBP5 homologs were closely related to AmelOBP2 and DmelOBP28a.

For phylogenetic analysis of CSPs, a phylogenetic tree was established based on 98 CSPs of eight species from different orders: Orthoptera (*L. migratoria*), Hemiptera *(A. pisum*), Diptera (*D. melanogaster*), Coleoptera (*T. castaneum*), Hymenoptera (*A. mellifera*), Lepidoptera (*S. exigua*) and Thysanoptera (*F. occidentalis* and *F. intonsa*) (Appendix A). Poor bootstrap support for deeper branches was found and most CSPs grouped according to species, which is consistent with the highly conserved characteristics of CSPs (Figure 5). Based on the phylogenetic tree, FoccCSP1/FintCSP1 homologs were closely related to ApisCSP5 and ApisCSP10. FoccCSP2/FintCSP2 and FoccCSP5/FintCSP5 homologs were sister groups and closely related to AmelCSP3 and TcasCSP19. FoccCSP3/FintCSP3 homologs were closely related to SexiCSP1. FoccCSP8 was a homolog of TcasOBP8, and were a sister group of FoccCSP4/FintCSP4 homologs. FoccCSP6/FintCSP6 homologs were closely related to TcasCSP5. FoccCSP7 was a homolog of ApisOBP1.

### 2.6. Expression Analyses of OBP and CSP Genes in Different Developmental Stages Based on Antennal Transcriptome and RT-qPCR

In the *F. occidentalis* antennal transcriptome, FoccOBP1, FoccOBP4, and FoccOBP6 exhibited high expression, whereas FoccOBP2, FoccOBP3, FoccOBP5, and FoccOBP7 exhibited low expression (Figure 6A). FoccCSP2, FoccCSP4, and FoccCSP7 exhibited high expression, whereas FoccCSP1, FoccCSP3, FoccCSP5, FoccCSP6, and FoccCSP8 exhibited low expression (Figure 6B). In *F. intonsa* antennal transcriptome, FintOBP1, FintOBP3, and FintOBP4 exhibited high expression, whereas FintOBP2, FintOBP5, and FintOBP6 exhibited low expression (Figure 6C). FintCSP2 and FintCSP4 exhibited high expression, whereas FintCSP1, FintCSP3, FintCSP5, and FintCSP6 exhibited low expression (Figure 6D).

The RT-qPCR was conducted to analyze the relative expression level of OBPs and CSPs at different developmental stages both in *F. occidentalis* and *F. intonsa*. In *F. occidentalis*, six OBPs were highly expressed in male adults, except for FoccOBP2 and FoccOBP7. FoccOBP2 was highly expressed at the larval stage, while FoccOBP7 was highly expressed at the pupal stage (Figure 7A). Among the eight CSP genes in *F. occidentalis*, FoccCSP1, FoccCSP2 and FoccCSP3 were highly expressed in male adults, while FoccCSP4, FoccCSP6, FoccCSP7, and FoccCSP8 were highly expressed at the larval stage; by contrast, FoccCSP5 was highly expressed at the pupa stage (Figure 7B). In *F. intonsa*, FintOBP1, FintOBP3, FintOBP4, and FintOBP6 were highly expressed in both female and male adults, while FintOBP5 was highly expressed in female adults (Figure 7C); FintOBP2 was evenly expressed in all the tested stages (Figure 7C). Among the six CSPs in *F. intonsa*, FintCSP1 and FintCSP2 were highly expressed in both female and male adults; FintCSP3 and FintCSP4 were highly expressed in all four stages; FintCSP5 was highly expressed in the laval and pupal stages, while FintCSP6 was highly expressed in larvae and female adults (Figure 7D).

### 2.7. Modeling of Three-Dimensional (3D) Structure and Molecular Docking of Ligands

The pocket parameters of OBPs and CSPs of *F. occidentalis* and *F. intonsa* calculated by DoGSiteScore were shown in Appendix A. Modeling results of OBP and CSP genes in *F. occidentalis* and *F. intonsa* were shown in Appendix A. The molecular docking results of OBPs and CSPs with aggregation pheromone compounds are shown in Table 3 and Figure 8.

Results from molecular docking showed that the active site residues and residues involved in H-bonding varied considerably both in OBPs and CSPs (Table 3). The most frequently used residues involved in H-bonding were TYR and LYS in OBPs, and SER and TYR in CSPs (Table 3). The H-bonding residues with the two aggregation pheromone compounds were not conserved between the two thrips species, even in OBP or CSP homologs. However, the residues that form hydrogen bonds with the two aggregation pheromone compounds were highly conserved in some OBPs and CSPs in both species (Table 3).

Among the seven OBPs from *F. occidentalis*, FoccOBP6, FoccOBP3 and FoccOBP4 showed lower binding energy with the major aggregation pheromone compound neryl (*S*)-2-methylbutanoate, with the binding energy −25.61, −25.06, and −24.94 kJ/mol, respectively, while FoccOBP7, FoccOBP3 and FoccOBP6 showed lower binding energy with the minor aggregation pheromone compound (*R*)-lavandulyl acetate, with the binding energy −24.02, −23.56 and −23.43 kJ/mol, respectively. Among the six OBPs from *F. intonsa*, FintOBP4, FintOBP6 and FintOBP1 showed lower binding energy with the major aggregation pheromone compound neryl (*S*)-2-methylbutanoate, with the binding energy −26.99, −25.82 and −25.48 kJ/mol, respectively, while FintOBP2, FintOBP4, and FintOBP6 showed lower binding energy with the minor aggregation pheromone compound (*R*)-lavandulyl acetate, with the binding energy −25.40, −22.84, and −22.38 kJ/mol, respectively. Consequently, orthologous pairs FoccOBP4/FintOBP4 and FoccOBP6/FintOBP6 might be involved in transporting the major aggregation pheromone compound neryl (*S*)-2-methylbutanoate (Figure 8A), and orthologous pair FoccOBP6/FintOBP6 might be involved in transporting the minor aggregation pheromone compound (*R*)-lavandulyl acetate (Figure 8B).

Among the eight CSPs from *F. occidentalis*, FoccCSP7, FoccCSP3 and FoccCSP2 showed lower binding energy with both aggregation pheromone compounds neryl (*S*)-2-methylbutanoate and (*R*)-lavandulyl acetate, with the binding energy −26.94, −25.73 and −24.85 kJ/mol for neryl (*S*)-2-methylbutanoate, and −22.22, −21.21 and −23.30 kJ/mol for (*R*)-lavandulyl acetate, respectively. Among the six CSPs from *F. intonsa*, FintCSP2, FintCSP1 and FintCSP6 showed lower binding energy with the major aggregation pheromone compound neryl (*S*)-2-methylbutanoate, with the binding energy −25.82, −23.43 and −22.59 kJ/mol, respectively, while FintCSP2, FintCSP3 and FintCSP5 showed lower binding energy with the minor aggregation pheromone compound (*R*)-lavandulyl acetate, with the binding energy −21.59, −21.59 and −21.55 kJ/mol, respectively. The binding energy of FoccCSP5 and FintCSP3 with neryl (*S*)-2-methylbutanoate was also low (−26.90 and −25.27 kJ/mol, respectively), yet no H-bond was formed. Consequently, the orthologous pair FoccCSP2/FintCSP2 might be also involved in transporting the major aggregation pheromone compound neryl (*S*)-2-methylbutanoate (Figure 8C), and orthologous pairs FoccCSP2/FintCSP2 and FoccCSP3/FintCSP3 might be involved in transporting the minor aggregation pheromone compound (*R*)-lavandulyl acetate (Figure 8D).

## 3. Discussion

The chemosensory system is crucial for insect behaviors. OBPs and CSPs that are mainly expressed in the olfactory organs may be involved in olfactory functions [14]. Thrips, such as *F. occidentalis* and *F. intonsa*, are among the most significant agricultural pests globally [44]. However, the chemosensory mechanism in thrips was not well explored, except for the report of a couple of OBPs and CSPs in *F. occidentalis* and *F. intonsa* in publications as well as in GenBank [51,52,53,54,55]. To expand our knowledge of the thrips chemosensory system, we conducted antennal transcriptome analysis of two thrips species, *F. occidentalis* and *F. intonsa*, and identified seven OBPs and eight CSPs in *F. occidentalis*, and six OBPs and six CSPs in *F. intonsa*. The number of OBPs and CSPs varies considerably across species [10,13,56]. The number of OBPs in insects ranged from 7 in *Megachile rotundata* to 109 in *Culex quinquefasciatus*, while the number of CSPs ranged from 4 in *D. melanogaster* to 70 in *L. migratoria* [13]. Unlike species from Diptera, Hymenoptera and Lepidoptera, where a large number of OBPs were identified [56], the number of OBPs in the two thrips species was very low. The low number of OBPs has also been reported in many species in Hemiptera, such as *Bemisia tabaci* (8 OBPs) [57], *Aphids gossypii* (9 OBPs) [58], *Myzus persicae* (9 OBPs) [59], *Nilaparvata lugens* (10 OBPs) [60] and *Sogatella furcifera* (12 OBPs) [61]. In case of CSPs, except for certain species (*L. migratoria* with 70 CSPs), the number of CSPs is relatively low across different insect species, and in many species, the number of CSPs is below 10 [13,62]. For instance, a low number of CSPs have been reported in *B. tabaci* (19 CSPs) [57], *A. gossypii* (9 CSPs) [58], *M. persicae* (9 CSPs) [59], *N. lugens* (11 CSPs) [60] and *S. furcifera* (9 CSPs) [63].

Since OBPs and CSPs play important roles as carriers for odors [13,14], it is generally assumed that the number of OBPs and CSPs should be related to the degree of diversity of odors. *F. occidentalis* and *F. intonsa* are polyphagous pests and would encounter diverse odors [34,35]. The relatively low number of OBPs and CSPs in these two thrips species has several possible explanations. First, the numbers of OBPs and CSPs identified in antennae in our study might be lower than the actual numbers, since it has been reported that some OBPs and/or CSPs are expressed in other tissues and organs (e.g., gut, fat body, legs, palps) instead of being limited to antennae [62,64]. Consequently, there might be additional OBPs and CSPs expressed in other body parts rather than antennae, which are not included in our study. Second, as with some species in Hemiptera, species in the Thysanoptera might in general have fewer OBPs and CSPs than species in other orders [59]. Like *Anopheles gambiae* [65], two or more OBPs might cooperatively bind ligands, thus the types of carrier proteins with different affinity abilities are increased. Third, the odorant receptors (ORs) interact with odors, initiate downstream signaling, and ultimately lead to behavioral responses, it is probable that one OBP or CSP could transport multiple odors to different ORs [4]. Therefore, the number of ORs, rather than OBPs and CSPs, might be related to the diversity of odor.

Based on comparative analysis, six OBPs and six CSPs in *F. intonsa* are orthologs of those in *F. occidentalis*. The similarity of OBP3, OBP5 and OBP6 in the two species was over 90%, while the similarity of CSP1, CSP2 and CSP4 in the two species was over 98%. One more OBP (FoccOBP7) and two more CSPs (FoccCSP7 and FoccCSP8) were identified in *F. occidentalis*. More soluble olfactory proteins identified from *F. occidentalis* might be due to the higher adaptability of this species [34,44]. *F. occidentalis* is a worldwide invasive species and has become one of the most significant horticultural pests in China [34]. It is a highly polyphagous insect with over 240 recorded host plants and can be exposed to a broad range of plant allelochemicals [44], which might result in more soluble olfactory proteins in this species. Shared aggregation pheromone compounds in *F. occidentalis* and *F. intonsa* [42,66] suggested that OBPs and CSPs with higher similarity, such as OBP3, OBP5, OBP6, CSP1, CSP2 and CSP4, might have a higher possibility of being involved in the recognition and transportation of the pheromone compounds neryl (*S*)-2-methylbutanoate and (*R*)-lavandulyl acetate.

Phylogenetic analysis among species from different orders revealed that all candidate *FoccOBPs/CSPs* and *FintOBPs/CSPs* showed extremely high homology in pairs, except for *FoccOBP7* and *FoccCSP7~8*. However, the OBPs or CSPs from the same thrips species did not cluster together but exhibited a wide divergence, indicating their capacity to cope with the diversity of semiochemicals in the environment [13]. Compared to other species, the OBPs from these two species are more closely related to OBPs from *A. pisum* and *S. exigua*, which share many host plants with thrips. This indicated that phylogenetically correlated OBPs probably constitute a functional cluster to separate and discriminate odors in a complicated environment [67]. By contrast, except for CSPs from *A. pisum* and *S. exigua*, CSPs from the two thrips species were also closely related to CSPs from *T. castaneum*, which is a huge difference between thrips. This might be due to the fact that, besides chemo-detection, CSPs also have other functions in insects [62].

The expression analyses based on antennal transcriptome data and RT-qPCR showed that most *FoccOBPs/CSPs* and *FintOBPs/CSPs* that were highly expressed during the adult stage, also showed high expression levels in antennae. This is consistent with a previous study, where one CSP gene in *F. occidentalis* was highly expressed both in the female stage and in antennae [54]. During the adult stage, host plant seeking, mating and reproduction are the most important behaviors for thrips [68,69], during which the detection of plant volatiles and thrips pheromones happens extensively. In both thrips species, male adults could release an aggregation pheromone which could be detected by both females and males [42,70]. Female adults also use plant volatiles to locate feeding and oviposition sites [69]. Consequently, these genes might be involved in the detection of plant volatiles and thrips aggregation pheromone. Expression analysis based on RT-qPCR indicated that some OBPs and CSPs (*FoccOBP2/7*, *FoccCSP2/4/5/6/7/8*, *FintOBP2* and *FintCSP3/4/5/6*) were highly expressed in the immature stage. Similar results have been reported in *F. occidentalis*, where higher expression levels were found for two OBPs and one CSP in the larval stage [51,53,55]. These genes might be involved in the detection of host volatiles [69], recognition of larvae-released alarm pheromone [71], or non-sensory functions [62]. Our results also indicated that some OBP and CSP genes expressed differently in thrips antennae and in different stages of *F. occidentalis* and *F. intonsa*. For instance, *FoccOBP3*, *FoccCSP1*, *FintOBP5*, *FintOBP6*, *FintCSP1* and *FintCSP6* showed low expression levels in the antennal transcriptome data, but were highly expressed in the adult stage. The expression level differences between antennal transcriptome data and RT-qPCR could be due to different expression levels of OBPs and CSPs in different body tissues [51,52]. Some OBP or CSP genes might exhibit higher expression levels in other body tissues compared to antennae. Tissue expression patterns of these genes need to be further explored.

Molecular docking has been widely used in structure-based drug design research [72] and recently has been frequently used for the functional prediction of insect OBPs and CSPs [73,74,75]. *F. occidentalis* and *F. intonsa* share two compounds, (*R*)-lavandulyl acetate and neryl (*S*)-2-methylbutanoate, as their aggregation pheromone compounds. Results from molecular docking showed that orthologous pairs *FoccOBP4/FintOBP4*, *FoccOBP6/FintOBP6* and *FoccCSP2/FintCSP2* had lower binding energy with major aggregation pheromone compound neryl (*S*)-2-methylbutanoate, while *FoccOBP6/FintOBP6*, *FoccCSP2/FintCSP2* and *FoccCSP3/FintCSP3* had lower binding energy with the minor aggregation pheromone compound (*R*)-lavandulyl acetate. The four orthologous pairs might be involved in binding and transporting thrips aggregation pheromone. Although molecular docking is a powerful method for ligand–protein interaction studies, it provides only static representations of dynamic olfaction systems, further robust experiments, such as fluorescence competitive binding, radioactively labeled ligand, RNAi, and site-directed mutagenesis should be carried out to confirm the function of these putative OBPs and CSPs in pheromone binding and transporting [28].

## 4. Materials and Methods

### 4.1. Insect Rearing

The colonies of *F. occidentalis* and *F. intonsa* were collected on *Cucumis melo* in a greenhouse at Chinese Academy of Agricultural Sciences, Beijing, and on *Cucumis sativus* at an experimental farm at Zhejiang Academy of Agricultural Sciences, Hangzhou, respectively. Cultures of the two thrips species were mass-reared on *Phaseolus vulgaris* bean pots in glass containers in separate climate rooms at room temperature of 27 ± 1 °C and relative humidity of 65–75% under an LD 16: 8 h photocycle.

### 4.2. Antennae Collection and RNA Extraction and Transcriptome Sequencing

To obtain known-age adults, pupae of each thrips species were collected from the cultures and reared individually until eclosion. One-day-old adults of *F. occidentalis* and *F. intonsa* were used for tissue collection. Antennae of *F. occidentalis* females (WFTF) and *F. intonsa* females (FTF) were dissected, immediately frozen in liquid nitrogen and stored at −80 ℃. Three biological replicates were included for each treatment, with around 100 thrips per replicate. Total RNA was isolated using Animal Tissue RNA Purification Kit (LC Science, Houston, TX) and sent to LC Sciences for transcriptome sequencing.

Since thrips are minute insects at 1–2 mm long, a trace amount of antennae tissue was collected, and consequently, SMART-seq (switching mechanism at 5′ end of the RNA transcript) was conducted for thrips antennae transcriptome analysis. Picogram amounts of total RNA were used to start SMARTer cDNA synthesis. The first-strand synthesis reaction was primed by the SMART CDS Primer. As SMARTScribe™ Reverse Transcriptase reached the 5′ end of the mRNA, a few additional nucleotides were added to the 3′ end of the cDNA under the enzyme’s terminal transferase activity. Then, the non-template nucleotide stretches paired with the SMARTer Oligonucleotide to create an extended template, enabling SMARTScribe RT to continue replicating to the end of the oligonucleotide. The resulting full-length, single-stranded cDNA contained the SMARTer Oligonucleotide complementary sequences as well as the complete 5′ end of the mRNA. Then, single-stranded cDNA was amplified by LD PCR to obtain enough dscDNA for library construction. Library construction began with fragmented cDNA, which was prepared by dsDNA Fragmentase (NEB, M0348S) at 37 °C for 30min. Blunt-end DNA fragments generation was conducted by combining the fill-in reactions and the exonuclease activity, and size selection was performed by using sample purification beads. Then, an A-base was added to the blunt ends of each strand, indexed Y adapters were ligated to the fragments, and PCR amplification was performed by using the ligated products. Finally, the paired-end sequencing was performed on an Illumina Novaseq™ 6000 (LC Sciences, Houston, TX, USA).

### 4.3. De Novo Assembly, Unigene Annotation and Functional Classification

Firstly, reads containing adaptor contamination, low-quality bases and undetermined bases were removed by using Cutadapt [76] and perl scripts in house. The sequence quality (the Q20, Q30 and GC-content of the clean data) was verified by using FastQC (http://www.bioinformatics.babraham.ac.uk/projects/fastqc/ (accessed on 11 October 2018)). Clean data of high quality were used for downstream analyses. De novo assembly of the transcriptome was performed by using Trinity 2.4.0 [77]. Transcripts were grouped into clusters based on shared sequence content, and the longest transcript was selected as the ‘gene’ sequence (aka unigene). Unigenes annotation were performed by searching against the Nr (NCBI non-redundant protein database, http://www.ncbi.nlm.nih.gov/ (accessed on 12 October 2018)), gene ontology (GO) (http://www.geneontology.org (accessed on 12 October 2018)), SwissProt (http://www.expasy.ch/sprot/ (accessed on 12 October 2018)), Kyoto Encyclopedia of Genes and Genomes (KEGG) (http://www.genome.jp/kegg/ (accessed on 12 October 2018)), eggNOG (http://eggnogdb.embl.de/ (accessed on 13 October 2018)) and Pfam (http://pfam.xfam.org/ (accessed on 13 October 2018)) databases using DIAMOND [78].

### 4.4. Identification of Putative OBP and CSP Genes

Putative OBP and CSP genes were identified based on the results of non-redundant protein (Nr), gene ontology (GO), SwissProt, and eggnog annotation from our antennal transcriptome dataset. OBP, CSP, odorant-binding protein, and chemosensory protein were used as keywords for annotated sequence screening. BLASTx and BLASTn searches (E-value < 10^−5^) were used for candidate chemosensory genes checking. BLASTX analysis with the nonredundant protein sequence (NR) database at Genbank (http://www.ncbi.nlm.nih.gov/ (accessed on 21 November 2021)) was used for the confirmation of putative chemosensory gene sequences. The conserved domains and open reading frames (ORFs) of these genes were predicted by the NCBI conserved domain search service (https://www.ncbi.nlm.nih.gov/Structure/cdd/wrpsb.cgi (accessed on 21 November 2021)) and ORF finder (https://www.ncbi.nlm.nih.gov/orffinder/ (accessed on 21 November 2021)), respectively. All candidate genes with complete ORF sequences were further validated by gene cloning and sequencing.

The putative N-terminal signal peptides prediction of OBPs and CSPs was conducted using the SignalP V4.1 (http://www.cbs.dtu.dk/services/SignalP/ (accessed on 21 November 2021)). Sequence alignment was performed using MAFFT (https://www.ebi.ac.uk/Tools/msa/mafft/ (accessed on 21 November 2021)) and the sequence identities of candidate OBP and CSP genes between *F. occidentalis* and *F. intonsa* were defined by sequence alignment on BLASTp (https://blast.ncbi.nlm.nih.gov/blast.cgi (accessed on 21 November 2021)).

### 4.5. Phylogenetic Analysis of OBPs and CSPs

The maximum likelihood phylogenetic trees of OBPs, as well as CSPs, were constructed with OBP and CSP amino acid sequences of *F. occidentalis*, *F. intonsa*, and species from 6 different orders: Orthoptera (*Locusta migratoria*), Hemiptera (*Acyrthosiphon pisum*), Diptera (*Drosophila melanogaster*), Coleoptera (*Tribolium castaneum*), Hymenoptera (*Apis mellifera*), and Lepidoptera (*Spodoptera exigua*) (Appendix A) using MEGA 6.0 software [79] with poison model and pairwise deletion of gaps. Bootstrap support of tree branches was assessed by re-sampling amino acid positions 1000 times. A circular phylogenetic tree was generated on Evolview (https://evolgenius.info//evolview-v2/#mytrees/ (accessed on 15 December 2021)).

### 4.6. Expression Analyses of OBP and CSP Genes in Different Developmental Stages Based on Antennal Transcriptome and RT-qPCR

The expression levels of the OBPs and CSPs identified in *F. occidentalis* and *F. intonsa* were analyzed using antennal transcriptome data. Gene expression levels of OBPs and CSPs in both species were estimated by RSEM and represented as FPKM values (fragments per kilobase of transcript per million mapped reads).

Real-time qPCR was used to compare the expression of OBPs and CSPs in different developmental stages. Second instars, pupae, females (1~3 days old) and males (1~3 days old) were collected and frozen in liquid nitrogen, and then stored at −80 °C until use. Each stage had 3 replicates. Total RNA was isolated using Trizol reagent (Invitrogen, Carlsbad, CA, USA). The RNA was quantified and checked for purity using NanoDrop ND 2000 Spectrophotometer and gel electrophoresis. The first single-strand cDNA was synthesized by using RevertAid First Strand cDNA Synthesis Kit (Thermo Fisher Scientific, Waltham, MA, USA). All procedures were conducted following the manufacturer’s instructions. The cDNA was stored at −20 °C.

Real-time qPCR was performed by using gene-specific primers (Appendix A), the iScript cDNA Synthesis Kit (BIO-RAD, Hercules, CA, USA), and the iTaq Universal SYBR Green Supermix (BIO-RAD, Hercules, CA, USA). β-actin gene was used as the reference gene for both species (Appendix A). All qPCR analyses included three technical replicates and three biological replicates.

### 4.7. Modeling of Three-Dimensional (3D) Structure and Molecular Docking of Ligands

The three-dimensional structure (3D) of OBP and CSP proteins was modeled through the SWISS-model portal (https://swissmodel.expasy.org/interactive (accessed on 4 May 2022)). The generated models were verified by Procheck [80]. The binding pocket parameters of OBPs and CSPs in the two species were calculated using DoGSiteScorer (https://proteins.plus (accessed on 4 May 2022)). Molecular docking of OBP and CSP proteins with aggregation pheromone compounds, neryl (*S*)-2-methylbutanoate and (*R*)-lavandulyl acetate, was carried out using Autodock 4.2 and AutoDock Tools 1.5.6. The 3D compound structures of neryl (*S*)-2-methylbutanoate and (*R*)-lavandulyl acetate were downloaded from PubChem (https://pubchem.ncbi.nlm.nih.gov (accessed on 4 May 2022)) and used as ligands. The default search parameters and docking parameters were used for molecular docking. Molecular visualization of docking results was obtained with PyMOL software.

## 5. Conclusions

Our study identified seven OBPs and eight CSPs in *F. occidentalis*, as well as six OBPs and six CSPs in *F. intonsa* based on antennal transcriptome analysis. High sequence identity was found in OBPs and CSPs between the two closely related species. Moreover, based on expression level and molecular docking, putative OBPs and CSPs that might be involved in binding and transporting the thrips aggregation pheromone were predicted. These results provide a fundamental basis for understanding the molecular mechanisms of pheromone reception in the two thrips species.

## Figures and Tables

**Figure 1 ijms-23-13900-f001:**
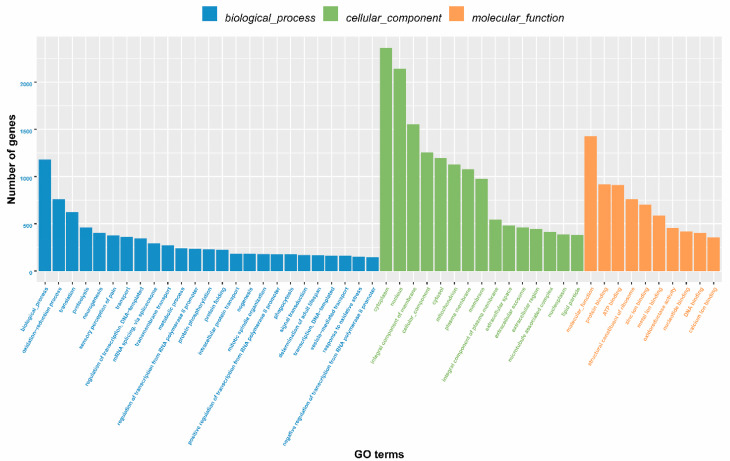
Gene ontology (GO) classification of *Frankliniella occidentalis* and *Frankliniella intonsa* unigenes.

**Figure 2 ijms-23-13900-f002:**
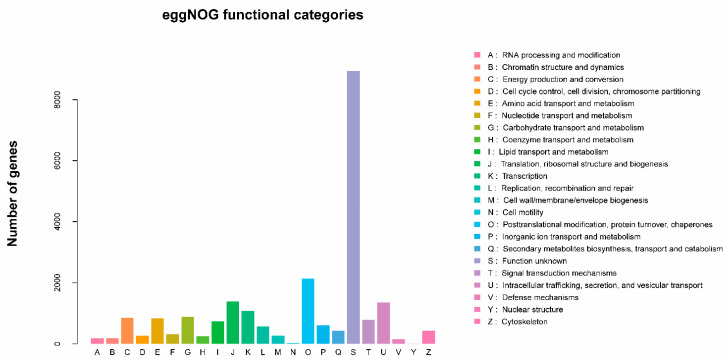
eggNOG functional categories of *Frankliniella occidentalis* and *Frankliniella intonsa* unigenes.

**Figure 3 ijms-23-13900-f003:**
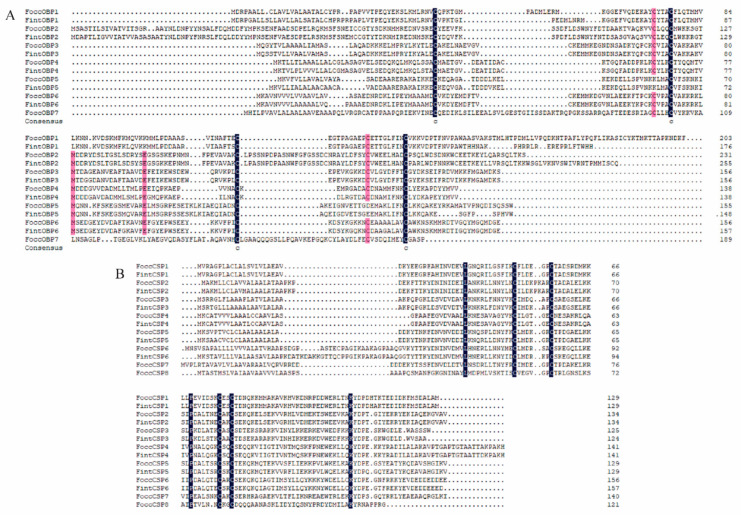
OBP (**A**) and CSP (**B**) amino acid sequence alignment in *Frankliniella occidentalis* and *Frankliniella intonsa*.

**Figure 4 ijms-23-13900-f004:**
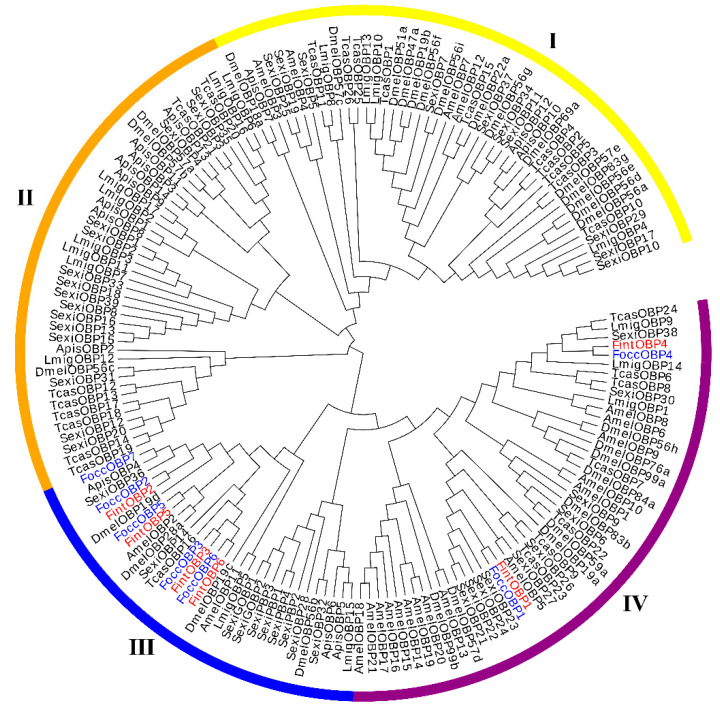
Maximum likelihood tree of OBPs from two *Frankliniella* species and representative species from six other insect orders. Species abbreviations are: Focc, *Frankliniella occidentalis* (Thysanoptera); Fint, *Frankliniella intonsa* (Thysanoptera); Apis, *Acyrthosiphon pisum* (Hemiptera); Amel, *Apis mellifera* (Hymenoptera); Dmel, *Drosophila melanogaster* (Diptera); Lmig, *Locusta migratoria* (Orthoptera); Sexi, *Spodoptera exigua* (Lepidoptera); Tcas, *Tribolium castaneum* (Coleoptera). Four sub-groups have been identified on the tree: I, Sub-group I; II, Sub-group II; III, Sub-group III; IV, Sub-group IV.

**Figure 5 ijms-23-13900-f005:**
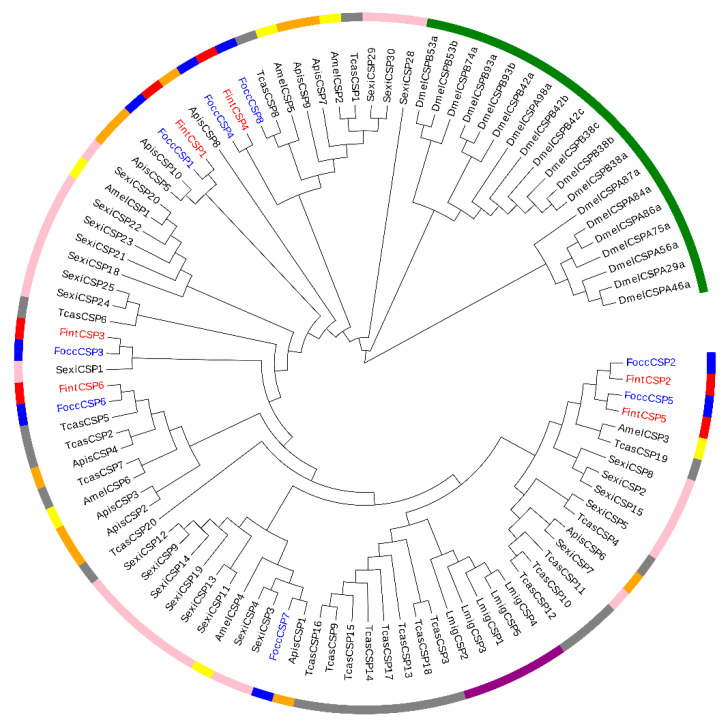
Maximum likelihood tree of CSPs from two *Frankliniella* species and representative species from six other insect orders. Species abbreviations are: Focc, in blue, *Frankliniella occidentalis* (Thysanoptera); Fint, in red, *Frankliniella intonsa* (Thysanoptera); Apis, in orange, *Acyrthosiphon pisum* (Hemiptera); Amel, in yellow, *Apis mellifera* (Hymenoptera); Dmel, in green, *Drosophila melanogaster* (Diptera); Lmig, in purple, *Locusta migratoria* (Orthoptera); Sexi, in pink, *Spodoptera exigua* (Lepidoptera); Tcas, in grey, *Tribolium castaneum* (Coleoptera).

**Figure 6 ijms-23-13900-f006:**
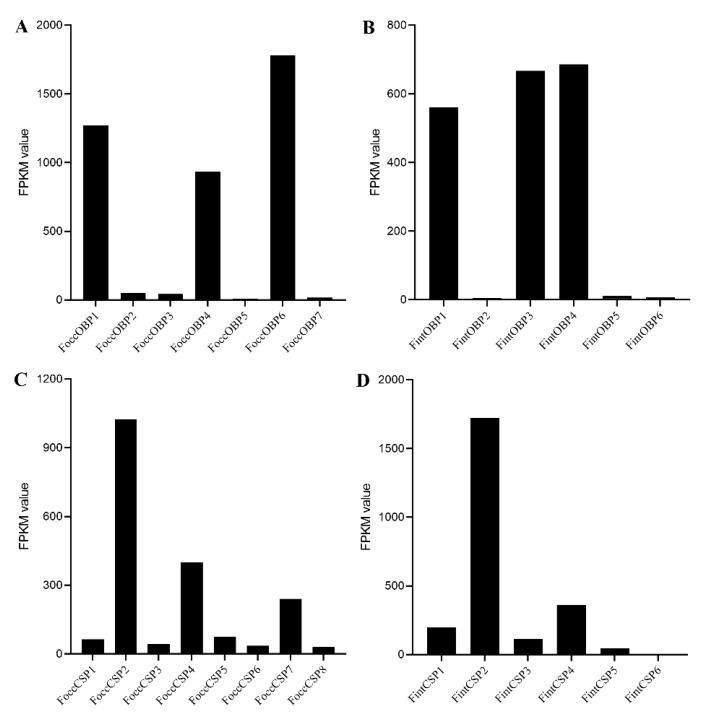
Expression level of OBPs (**A**) and CSPs (**C**) in *Frankliniella occidentalis*, and OBPs (**B**) and CSPs (**D**) in *Frankliniella intonsa* based on the antennal transcriptome. FPKM value: fragments per kilobase of transcript per million mapped reads.

**Figure 7 ijms-23-13900-f007:**
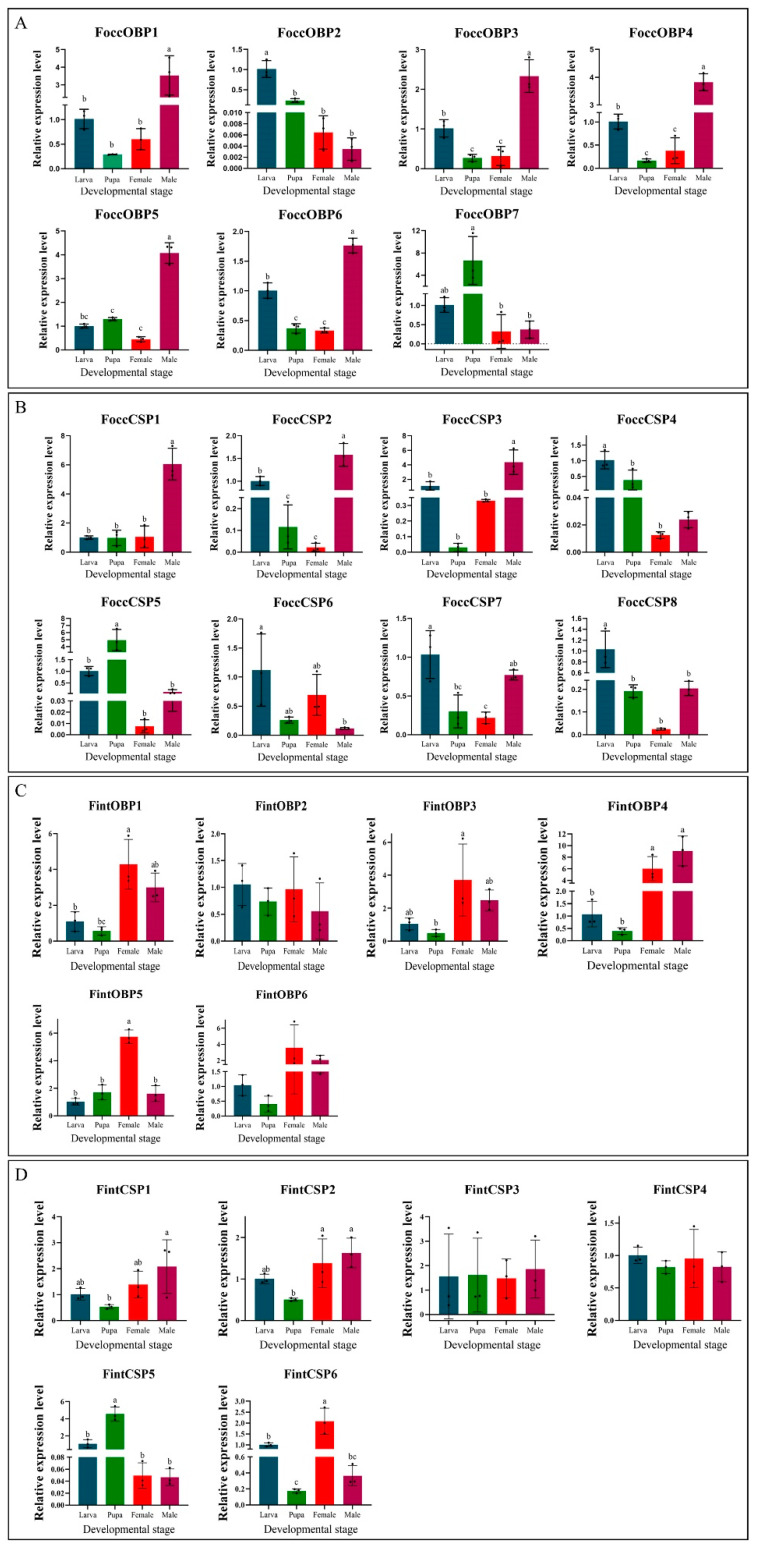
Expression analyses of OBP (**A**) and CSP (**B**) genes of *Frankliniella occidentalis*, and OBP (**C**) and CSP (**D**) genes of *Frankliniella intonsa* in different developmental stages by qRT-PCR. The bar represents the standard error, and the different letters above each bar indicate significant differences (*p* < 0.05).

**Figure 8 ijms-23-13900-f008:**
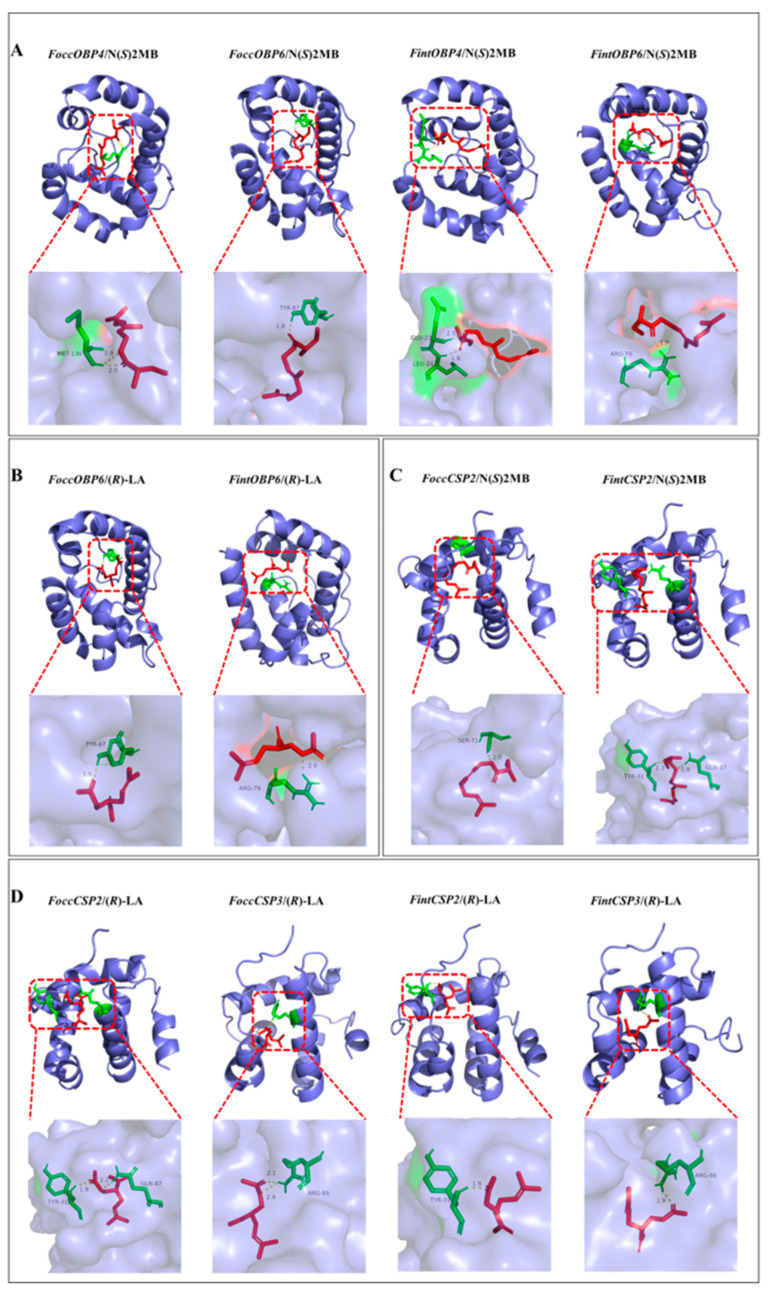
Three-dimensional models of OBPs/CSPs and aggregation pheromone compounds interactions in *Frankliniella occidentalis* and *Frankliniella intonsa*. N(*S*)2MB: neryl (*S*)-2-methylbutanoate; (*R*)-LA: (*R*)-lavandulyl acetate. (**A**). Interactions between N(*S*)2MB and FoccOBP4/FoccOBP6/FintOBP4/FintOBP6; (**B**). Interactions between (*R*)-LA and FoccOBP6/ FintOBP6; (**C**). Interactions between N(*S*)2MB and FoccCSP2/FintCSP2; (**D**). Interactions between (*R*)-LA and FoccCSP2/FoccCSP3/FintCSP2/FintCSP3.

**Table 1 ijms-23-13900-t001:** List of odorant-binding proteins (OBPs) and chemosensory proteins (CSPs) in *Frankliniella occidentalis* and *Frankliniella intonsa*.

Species	Gene Name	Signal Peptide (aa)	Amino Acids (aa)	ORF (bp)	Full ORF	Accession No.
*Frankliniella occidentalis*	*FoccOBP1*	22	203	610	No	OP380934
	*FoccOBP2*	19	231	696	Yes	OP380936
	*FoccOBP3*	21	156	471	Yes	OP380938
	*FoccOBP4*	20	138	417	Yes	OP380940
	*FoccOBP5*	20	155	465	Yes	OP380942
	*FoccOBP6*	20	156	471	Yes	OP380944
	*FoccOBP7*	19	189	570	Yes	OP380946
	*FoccCSP1*	20	129	390	Yes	OP380946
	*FoccCSP2*	19	134	405	Yes	OP380949
	*FoccCSP3*	22	125	375	Yes	OP380951
	*FoccCSP4*	20	142	426	Yes	OP380953
	*FoccCSP5*	20	129	390	Yes	OP380955
	*FoccCSP6*	22	156	471	Yes	OP380957
	*FoccCSP7*	19	140	423	Yes	OP380959
	*FoccCSP8*	27	121	366	Yes	OP380960
*Frankliniella intonsa*	*FintOBP1*	23	174	529	No	OP380935
	*FintOBP2*	20	255	768	Yes	OP380937
	*FintOBP3*	21	156	471	Yes	OP380939
	*FintOBP4*	20	138	417	Yes	OP380941
	*FintOBP5*	19	148	444	Yes	OP380943
	*FintOBP6*	21	157	474	Yes	OP380945
	*FintCSP1*	20	129	390	Yes	OP380948
	*FintCSP2*	19	135	405	Yes	OP380950
	*FintCSP3*	22	124	375	Yes	OP380952
	*FintCSP4*	20	142	426	Yes	OP380954
	*FintCSP5*	20	129	390	Yes	OP380956
	*FintCSP6*	19	157	474	Yes	OP380958

**Table 2 ijms-23-13900-t002:** Alignment of OBP and CSP protein sequences in *Frankliniella occidentalis* and *Frankliniella intonsa*.

Gene Name		Total Score	Query Cover	E-Value	Per. Identify%
*FoccOBP1*	*FintOBP1*	249	75%	2 × 10^−90^	89.47
*FoccOBP2*	*FintOBP2*	384	99%	2 × 10^−141^	76.72
*FoccOBP3*	*FintOBP3*	296	100%	8 × 10^−110^	91.72
*FoccOBP4*	*FintOBP4*	231	100%	1 × 10^−84^	84.17
*FoccOBP5*	*FintOBP5*	218	75%	5 × 10^−79^	91.38
*FoccOBP6*	*FintOBP6*	265	87%	1 × 10^−97^	93.43
*FoccCSP1*	*FintCSP1*	265	100%	3 × 10^−98^	98.46
*FoccCSP2*	*FintCSP2*	267	100%	5 × 10^−99^	98.52
*FoccCSP3*	*FintCSP3*	189	99%	1 × 10^−68^	83.87
*FoccCSP4*	*FintCSP4*	283	100%	3 × 10^−105^	98.59
*FoccCSP5*	*FintCSP5*	212	84%	2 × 10^−77^	91.82

**Table 3 ijms-23-13900-t003:** Docking score and molecular docking results of OBPs and CSPs with aggregation pheromone compounds of *Frankliniella occidentalis* and *Frankliniella intonsa*.

Gene Name	N(*S*)2MB		(*R*)-LA	
	Mean Binding Energy (kJ/mol)	Residues Forming H-Bond with Ligand	Mean Binding Energy (kJ/mol)	Residues Forming H-Bond with Ligand
*FoccOBP1*	−21.80	TYR74	−21.13	TYR74
*FoccOBP2*	−22.89	TYR143	−21.63	VAL110
*FoccOBP3*	−25.06	LYS25, MET28	−23.56	LYS25
*FoccOBP4*	−24.94	MET136	−23.05	MET136
*FoccOBP5*	−23.35	LYS29	−20.17	PHE76
*FoccOBP6*	−25.61	TYR87	−23.43	TYR87
*FoccOBP7*	−24.73	LEU169	−24.02	LEU169
*FintOBP1*	−25.48	VAL150	−22.22	TYR38
*FintOBP2*	−23.30	TRP124	−25.40	TRP124
*FintOBP3*	−22.89	GLY135	−19.54	TYR31
*FintOBP4*	−26.99	GLU23, LEU24	−22.84	GLU23
*FintOBP5*	−22.38	LYS31	−18.03	ASN70
*FintOBP6*	−25.82	ARG79	−22.38	ARG79
*FoccCSP1*	−23.39		−18.79	THR106
*FoccCSP2*	−24.85	SER71	−21.21	TYR31, GLN87
*FoccCSP3*	−25.73	ARG86	−23.30	ARG86
*FoccCSP4*	−20.42	SER57	−20.67	PHE26, SER57
*FoccCSP5*	−26.90		−19.62	SER66
*FoccCSP6*	−23.72	SER66	−20.67	ARG74
*FoccCSP7*	−26.94	TYR129	−22.22	TYR129
*FoccCSP8*	−19.96	ASN91	−19.54	TYR107
*FintCSP1*	−23.43	THR106	−20.71	TYR31
*FintCSP2*	−25.82	GLN87, TYR31	−21.59	TYR31
*FintCSP3*	−25.27		−21.59	ARG86
*FintCSP4*	−19.50	VAL31	−19.37	SER57
*FintCSP5*	−22.59	THR86	−21.55	ARG74
*FintCSP6*	−21.63	GLN111	−21.21	GLN111

N(*S*)2MB: neryl (*S*)-2-methylbutanoate; (*R*)-LA: (*R*)-lavandulyl acetate.

## Data Availability

The data presented in this study are available in the article.

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
