# Peer review of "Comparison and Functional Analysis of Odorant-Binding Proteins and Chemosensory Proteins in Two Closely Related Thrips Species, *Frankliniella occidentalis* and *Frankliniella intonsa* (Thysanoptera: Thripidae) Based on Antennal Transcriptome Analysis"

_ijms, 2022, doi:10.3390/ijms232213900_

Round 1

Reviewer 1 Report

The overall paper theme is interesting but the docking part is not convincing. Indeed, the difference in binding energies are too small to conclude whether either protein has a transport role (or not).

In addition, a lot of information are missing: the choice of the binding zone used for the docking studies, the conservation (or not) of the active site residues from a protein to another (and between the two insect species), the conservation (or not) of residues involved in H-bonding,….

Figure 8 brings nothing interesting. The active sites should be in the same orientation for each protein. And the same residues should be pictured in all the schemes, so that we can compare the binding modes.

In the introduction, no structural information is mentioned (which is known about the active site of OBP and CSP? Are there already insect OBP and CSP in the PDB databank?....). This is essential for the docking analyses.

Author Response

Comment 1: The overall paper theme is interesting but the docking part is not convincing. Indeed, the difference in binding energies are too small to conclude whether either protein has a transport role (or not).

Response: We added discussion about this issue in the revised manuscript. Please see lines 435-441 in the revised manuscript.

Comment 2: In addition, a lot of information are missing: the choice of the binding zone used for the docking studies, the conservation (or not) of the active site residues from a protein to another (and between the two insect species), the conservation (or not) of residues involved in H-bonding,….

Response: The method and results of the binding pocket prediction were added, please see lines 565-566, and supplementary Table S4 in the revised manuscript. The discussion of conservation of the active site residues and residues involved in H-bonding were added. Please see lines 280-290 in the manuscript.

Comment 3: Figure 8 brings nothing interesting. The active sites should be in the same orientation for each protein. And the same residues should be pictured in all the schemes, so that we can compare the binding modes.

Response: Figure 8 was reprepared according to your suggestion. Please see Figure 8 in the revised manuscript.

Comment 4: In the introduction, no structural information is mentioned (which is known about the active site of OBP and CSP? Are there already insect OBP and CSP in the PDB databank?....). This is essential for the docking analyses.

Response: We have added the structural information of OBPs and CSPs in the revised manuscript. Please see lines 64-79 in the revised manuscript.

Reviewer 2 Report

The article by Li et al. (ijms-1946198) “Comparison and functional analysis of odorant-binding proteins and chemosensory proteins in two closely related thrips species, Frankliniella occidentalis and Frankliniella intonsa (Thysanoptera: Thripidae) based on antennal transcriptome analysis” focuses on chemosensory genes and olfactory mechanisms in these two species for further understanding their molecular mechanisms of pheromone reception.

This study is a continuation of previous work from the authors (Li et al. (2019) [38]). The manuscript is very well-written and easy to understand. The experiments were adequately designed and results clearly presented and interpreted. I found very few comments to improve the study here presented, which has been submitted in a high-quality background.

-     Please, improve resolution for Figures 1, 3-5 and 7

-     I suggest the authors include more references throughout the manuscript, concretely in the introduction and conclusion sections.

-     The discussion should be extended and cover all results presented in the manuscript. Additionally, I did not find this section appropriately supported along by the bibliography.

Line 63-63: Please, indicate the reference of the study.

Author Response

The article by Li et al. (ijms-1946198) “Comparison and functional analysis of odorant-binding proteins and chemosensory proteins in two closely related thrips species, Frankliniella occidentalis and Frankliniella intonsa (Thysanoptera: Thripidae) based on antennal transcriptome analysis” focuses on chemosensory genes and olfactory mechanisms in these two species for further understanding their molecular mechanisms of pheromone reception. This study is a continuation of previous work from the authors (Li et al. (2019) [38]). The manuscript is very well-written and easy to understand. The experiments were adequately designed and results clearly presented and interpreted. I found very few comments to improve the study here presented, which has been submitted in a high-quality background.

Comment 1: Please, improve resolution for Figures 1, 3-5 and 7

Response: We have improved the resolution for Figures 1, 3-5 and 7. Please see attached figures in the system.

Comment 2: I suggest the authors include more references throughout the manuscript, concretely in the introduction and conclusion sections.

Response: Done. We have added more references throughout the manuscript.

Comment 3: The discussion should be extended and cover all results presented in the manuscript. Additionally, I did not find this section appropriately supported along by the bibliography.

Response: Thank you for your comment. We have extended the discussion section and added more references.

Comment 4: Line 63-63: Please, indicate the reference of the study.

Response: Done.

Round 2

Reviewer 1 Report

The authors have correctly answered to the reviewer's suggestions/remarks. The paper is now suitable for publication.

Author Response

Thank you for your comments.